# Marine Stewardship Council Certification in Finland and Russia: Global Standards and Local Practices

Svetlana Tulaeva [1,*], Maria Tysiachniouk [1], Minna Pappila [2] and Minni Tynkkynen [3]

[1] Department of Geographical and Historical Studies, University of Eastern Finland, PL 111, 80101 Joensuu, Finland
[2] Finnish Environment Institute, Latokartanonkaari 11, 00790 Helsinki, Finland
[3] Faculty of Management and Business, Tampere University, Kanslerinrinne 1, 33100 Tampere, Finland
* Correspondence: svetta05@gmail.com

**Abstract:** The state of seafood resources around the world has been declining for the last 50 years. There are multiple global, regional, and national regulatory arrangements that make an effort to revert this situation. The Marine Stewardship Council (MSC) is a voluntary global instrument, believed to foster sustainability in commercial fishing practices. This paper analyzes the institutionalization of MSC in Finland and Russia, and highlights how MSC as a global standard adapts to the different local contexts. It also shows which other global regulatory arrangements contribute to regulating fish production and what are the specifics of interaction between them. For the analysis of the MSC scheme, this paper uses the governance generating network (GGN) theory, which has been widely applied to the research on the FSC forest certification scheme and oil sector. The GGN lens helps to analyze the generative capacity of multiple global regulatory instruments including MSC in the Baltic Sea (Finland) and the Barents Sea (Russia). Qualitative methodology, such as semi-structured interviews with the same interview guide, document analysis, and participant observations were used in both Finland and Russia. We show that several GGNs are contributing to fishing regulations, e.g., the implementation of MSC in both countries is facilitated by multiple international organizations and conventions, which were signed prior to the creation of the MSC scheme. The limited added value of MSC certification is observed in both Finland and Russia: MSC ensures economic stability of certified companies and contributes to biodiversity conservation.

**Keywords:** Marine Stewardship Council; certification; MSC; marine governance; governance generating networks; GGN theory; market-driven regulation; sustainability of food production; sustainable fishing

## 1. Introduction

The state of marine fish resources has been declining for the last 50 years, and the percentage of stocks fished at biologically unsustainable levels has increased to 34.2% in 2017 [1]. The largely inefficient governmental policy development and thus mitigation of the collapse of international marine fisheries have worked as incentives for measures that are more global in scale. These market-based strategies, such as ecolabeling, have their aim at creating awareness and educating consumers on the environmental and social effects of the purchases they make daily, advocating for a change towards a more sustainable industry and directing consumers' purchasing behavior [2–4]. An example of market-driven governance in the sphere of marine resources is the Marine Stewardship Council voluntary certification (MSC). The Marine Stewardship Council (MSC) was launched to help tackle overfishing and to drive improvement in fisheries management. MSC originated from the cooperation of two global actors, World Wildlife Fund (WWF) and Unilever in 1997 [5]. WWF was willing to intervene in the market through amplifying adoption of the certified products in the major supermarket chains and retailers [6], whereas Unilever aimed for strengthening its sustainable image and assuring continuous supply [4–6]. To be exact, there are two types of standards underneath the same brand name. The MSC

Fisheries Standard founded in 1998 is based on the principles listed above, and the MSC Chain of Custody (CoC) Standard that came to life in 2001 [6], as a means of verifying traceability of the MSC products from "ocean to plate" [7].

According to MSC, a sustainably ranked fishery can be outlined by their practices condensed as the following principles: (1) sustainable fish stocks: a fish population's sustainability shall not be compromised by the fishing activities; (2) minimizing environmental impact: the ecosystem and its structure, productivity, diversity or function should not suffer harm by the fishing operations; and (3) effective management: the fishing company must abide by all local, national, and international laws, as well as have an environmental management system in place [8].

MSC strives to reward environmentally sustainable fisheries and their practices while also maintaining a capacity building program to support the attainment of these objectives in the long run. It is essential in the context of a lacking or insufficient government intervention, to limit, for example, the unintended bycatch caused by damaging techniques that are a threat to biodiversity [4,5]. The label aims to provide traders, retailers, as well as consumers sufficient and reliable information on practices of production, processing, and trading [4].

Today, the total number of MSC-engaged (meaning being either certified, suspended or in MSC assessment) catch has reached 16 million tons, with MSC being the biggest seafood certifier. Its 409 certified fisheries (of which 22 were suspended) from 53 countries accounted for 17.4% of global marine catch in 2020 [9,10]. As of fall 2021, there were more than 20,000 products holding the MSC label, and 46,205 chain of custody certificate holding sites. According to the statistics of 2016 and 2017, most of the certified fisheries, retailers, processors, and customers (over 90%) were coming from Northwestern Europe and North America—perhaps due to the comparatively active NGO basis. In 2019–2020, of all wild marine catch, 15% was MSC-certified, and based on seafood volume, the organization is one of the leading environmental standards [1,11,12].

Despite the general increase of MSC influence in the world, its role in the sustainable governance of marine resources in different countries is widely discussed. There are parties who support MSC as a label fostering sustainability and those who perceive it as useless in governance and consequently in conservation of marine environment. Those actors feel that international conventions and national legislation is sufficient in regulation of fishing practices [6]. Aspects such as methodologies of inconsistent certification, performance indicators leading to non-consecutive interpretations of criteria and principles, alleged needless focus on environmental improvements if management practices are in place, and MSC's contribution to problem-solving being modest at best have been criticized throughout the years [5,7]. Also, the fisheries currently certified by the organization were neither considered being in the most environmentally vulnerable state, nor representing smaller scale fisheries in the developing countries by experts, but instead consist of markets where an additional price can be paid for ecolabeling the fish [8]. Providing the necessary environmental data that constitutes the basis for a Fisheries Standard certification process in addition to the technical and financial requirements can turn out to be a problem for some small-scale operators as well. MSC was also criticized because the same assessment methodology for certifying fisheries was used in developing and developed countries [4,8]. Some questions are connected with the complexity of governing a global certification scheme and its supply chains [6].

Controversial evaluations and effects of the MSC impact on sustainable fisheries have been reflected in the academic debate [13–16]. Several main directions of changes under the influence of MSC are identified in the literature: (a) structural effects, such as changes in markets or in power relations [17–20]; (b) cognitive effects such as construction or promotion of certain discursive frameworks [21]; and (c) regulatory effects, such as the influence of transnational rules and standards on government regulation processes [22,23]. Researchers use different approaches to explain the stability/instability of the MSC certification. Some researchers focus on the opportunities and limitations of market instruments and market

competition to increase sustainability in natural resource extraction [17,24,25]. These papers highlighted the effectiveness of ecolabels, including MSC [17,26,27]. There is also research that looks at stakeholder attitudes and recognition of the label in the supply chain [19]. Another part of the research focuses on analyzing the embeddedness of MSC in global governance. It shows how the layering of different international regulations can strengthen or weaken marine certification [28,29]. Some researchers identified and measured environmental impact [30,31]. Another group of researchers focuses on the interaction of maritime certification with national regulation [32,33]. They demonstrate that MSC can influence the adoption of stronger national regulation for the development of sustainable fisheries. However, in some cases, MSC may allow government regulation to be "covered" by less stringent rules and standards and used as an excuse for inaction [33].

Our approach is slightly different. In this paper we will look at MSC as a global governance instrument with its special characteristics and compare the institutionalization of MSC in Finland and Russia. We will look at the interplay of different global regulatory instruments. We will analyze and assess whether there is added value in MSC certification to national and international regulatory tools in either country. In our comparative analysis of MSC in Finland and Russia, we look at the governance of this certification scheme and analyze relationships with actors that support/facilitate MSC certification in both countries. We explain to what extent interstate agreements and International NGOs can influence MSC certification and whether there is added value in MSC in regulation of fishing.

Therefore, the paper will answer the following research questions:

1. How do global standards adapt to the local context within MSC?
2. What factors influence the specifics of the institutionalization of MSC standards in each country?
3. What is the added value of MSC to other global regulatory arrangements?

We start the paper with a literature review on the history of MSC and its standards. Next, we describe how we use the Governance Generating Networks (GGNs) concept as a theoretical lens, explaining why it is helpful for understanding global/local interplay of the actors involved in market regulation [34,35]. In the following section, we describe our qualitative research methodology, and its capacity to reconstruct the specific features of MSC implementation in both countries. In the results section, we describe each of the case studies, for example, MSC in Finland and MSC in Russia. We compare cases in the discussion section and highlight the added value of our research to the field of governance of natural resources in conclusions. At the end we reflect on perspectives for future research.

This paper contributes to the understanding of global governance with voluntary certification as a tool. We analyzed different global and national regulatory instruments and highlighted the added value of MSC as a newly introduced regulatory institution. GGN was widely used by many authors to explain the operation of the Forest Stewardship Council (FSC) and governance of oil. This paper is the first in using this explanatory framework in marine governance. The paper looks at MSC-GGN structure and complex agency. GGN allows us to look at both the agency and structures of networks that are linking global and local scales, transferring the standard developed in the node to the sites of implementation. The innovative approach of this paper is in the analysis of different GGNs contributing to the same site of implementation. They turned out to be both drivers and promoters of MSC-GGN, involving those that were initiated by state actors and interstate institutions. This paper is the first in analyzing how GGNs in marine governance intersect with one another, what is their facilitating role, how they contribute to regulation of fishing, and what is the added value of MSC requirements. This approach makes this paper state of the art.

## 2. Theoretical Framework

The GGNs concept is useful for looking at transnational governance of natural resources in the globalized world [34–38]. GGNs develop and implement global standards, rules, norms or recommendations. It consists of three structural elements: the nodes of

global governance design, the forums of negotiation, and the sites of implementation [37,39]. GGN involves centers in transnational spaces called nodes and territories in particular places—sites of implementation. Forums of negotiation may be transnational, national, local, and across scales. There are many networks with transnational nodes in which actors are engaged in developing global regulatory standards. However, not all of them become effective GGNs with global standards that foster institutional changes around the world in concrete sites of implementation [40].

In the field of natural resource management, the FSC-GGN was one of the first, followed by MSC, Aquaculture Stewardship Council (ASC), and Tourism Stewardship Council (TSC) [8].

GGNs may be driven by NGOs, which are transnational corporations (TNCs) of supra-national institutions. Relationships between different GGNs differ, and some have familial relationships. For example, those which were originally initiated by WWF, such as FSC, MSC, ASC, and TSC. Other GGNs, such as FSC and PEFC, may compete for the sites of implementation, for example, different kinds of forest certification [41,42].

Certification GGNs may be supported by NGOs, governmental purchasing programs, retailers, and investment banks. All these actors-promoters foster legitimacy of the certification scheme in both transnational spaces and sites of implementation [42]. When joining a certification GGN, companies transfer the standards developed in the transnational nodes into concrete practices in terrestrial or aquatic sites of implementation.

The certification GGNs operating in different production sectors do not compete. On the contrary, they build alliances to resolve various issues and to develop a joint vision of standardization and labeling. Thus, ISEAL, since 1999, unites different certification GGNs, such as FSC, MSC, Fair-trade Labeling Organization (FLO), the International Federation of Organic Agriculture Movement (IFOAM), the International Organic Accreditation Service (IOAS), and many others. In this way ISEAL became an umbrella-GGN, regulating 'sensitive' niche markets and fostering sustainability globally. On behalf of its members, ISEAL negotiates at large forums, such as WTO, Organization for Economic Cooperation, and Development (OECD) or UN Committee on Trade and Development (UNCTAD), putting pressure on the world's policies and promoting market niches of sustainably certified products. Therefore, the ISEAL alliance promotes 'sensitive' markets in several sectors of the economy, such as forestry, fishing, fair trade, organic agriculture, and others [43].

In fact, certification schemes require compliance with existing international conventions in a particular field, for example, FSC requires compliance with CBD and ILO. Global and European governmental agreements, ministerial processes, and national legislation all approach the same issues and their requirements interlace [41,42]. In this way they make it easier for companies engaging in certification to comply with the standards. Vice versa, certification GGNs help enforce compliance with supra-state and state agreements. Such facilitation can happen because the design of state-driven and private-driven networks is directed towards solving the same issues at the sites of implementation [40].

MSC-GGN was promoted by WWF, is part and parcel of the ISEAL alliance, adopted the FAO Code of Conduct for responsible fisheries, and was supported by Global Sea Food Initiative [44,45]. Each of these global entities represent facilitating GGNs that helped the institutionalization of MSC around the world. The Total Allowable Catch (TAC) for each fish species is determined by another facilitating GGN, the International Council for the Exploration of the Sea (ICES), composed of scientists, which constantly organizes forums of negotiation and coordinates with research monitoring institutions in particular countries [46].

In addition to global GGNs regulating fish stocks, there are region-specific regulatory bodies. In Finland fishing is highly regulated by the EU Fisheries Policy, which requires compliance with requirements comparable with MSC. In Russia, the Russian-Norwegian Cooperation Agreement plays an important role. In addition, The North East Atlantic Fisheries Commission (NEAFC) plays a role as it regulates TAC in the Arctic and Atlantic Oceans [47]. In this paper we will assess the role of MSC-GGN in regulating fishing

practices at the sites of implementation in Russia and Finland and evaluate whether it has an added value in marine ecosystems preservation (see Figure 1).

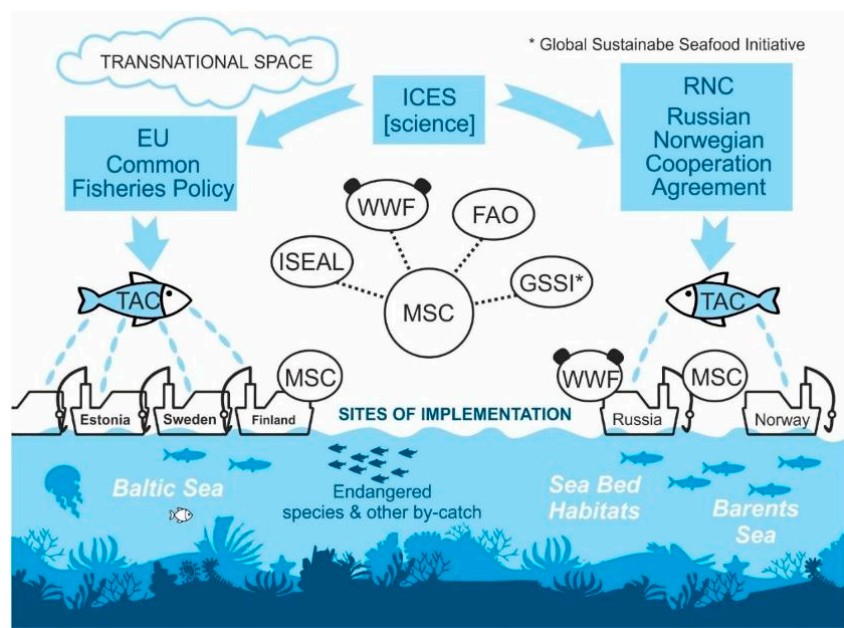

**Figure 1.** GGNs regulating fishing in the Baltic and Barents Sea.

The institutionalization of MSC-GGN in the world has occurred in different ways. Individual governments were initially skeptical of the MSC-GGN and tried to develop their own labeling systems and guidelines for sustainable fisheries. There were efforts to create regional GGNs. For example, in 2001 the Nordic Council of Ministers established a Nordic Technical Working Group on Fisheries Labeling Criteria. There were efforts to create national private fisheries regulation schemes. For example, in Sweden, government authorities and businesses initially rejected the MSC program and encouraged the development of an alternative ecolabel, which was created in 2004 by the Swedish non-governmental organization Organic Production Control Association (KRAV) [8,48]. The Icelandic government and industry representatives used the 2005 FAO guidelines to develop their own ecolabeling program in 2007. They also developed a guide to sustainable fisheries in Iceland [30].

Experts identify several key factors influencing the specifics of the institutionalization of MSC-GGN in a country. First, it is the focus of companies in international markets given the associated economic benefits from exporting green-label products. If product certification opens the way for national companies to international markets, then states turn out to be more supportive of non-state institutions. The governments of the seafood-producing countries are interested in promoting the commercial development of the seafood industries operating within their jurisdiction, in particular in expanding market opportunities for domestic export-oriented industries. For example, many coastal states depend on MSC for the regular export of seafood.

Second, the features of the spread of non-state GGNs are determined by the specifics of the resource. For example, marine fish stocks are shared resources managed by governments through international, regional, and domestic governance regimes. This determines the importance of international structures in this area, as well as the involvement of national governments in international processes. The higher the degree of internationalization of resource management, the easier it is to institutionalize non-state standards in the country. The active interaction of state authorities in the discussion of these issues in the international arena increases their readiness to implement international standards in the country. As it was explained earlier, since fish is subject to a large number of international agreements, several facilitating GGNs enable MSC-GGN.

Third, the implementation of MSC depends on national regulation of natural resources, including rules and management structures. The fewer the differences in the requirements of different regulators, the easier it is to institutionalize the new rules. In some cases, governments use international standards as "benchmarks" for "best practice" in developing their current policies and improving governance. International standards are expected to help improve management quality, such as increasing product legality. In addition, international assessments of the state of natural resources in the country can help to improve the image of the state in the international arena, or, conversely, undermine the legitimacy of the existing government regime. Fourth, it is the existence of a civil society that is able to be part of MSC-GGN, to monitor the implementation of the announced promises by companies and organize information campaigns and boycotts against violators. Fifth, it is the level of integration of companies and the presence of strong associations joining the MSC-GGN for facilitating the certification process [19].

Thus, institutionalization of global standards at the sites of implementation is accompanied by a number of transformations caused by the peculiarities of adaptation to the local context, which we will highlight in this paper. Analyzing MSC certification via the GGN lens, we will compare the added value of MSC in Finland and Russia. We will highlight how global standards adapt to the local context within MSC-GGN. We will analyze the dynamic interplay of global and regional GGNs that are focusing on governance of the Baltic and Barents Sea. A significant part of our analysis is dedicated to the global/local interplay of actors involved in governance of marine resources in Finland and Russia. We will scrutinize which factors influence the interaction of global and local actors during institutionalization and implementation of MSC.

## 3. Materials and Methods

Qualitative methodology was relevant for the phenomenological study of the institutionalization and implementation of MSC [49–52]. The main research strategy was a comprehensive study and comparison of the governance arrangements involving MSC and other global regulatory instruments in two neighboring countries, Finland and Russia. In Finland, in the Baltic Sea, and in Russia, in the Barents Sea. MSC standards are global, yet both international agreements and the state system for governance of marine resources differ in these two countries, which makes a fruitful starting point for our comparison. The configuration of international actors involved in the international governance of the extraction of marine resources in these two regions was also different. Comparative analysis made it possible to analyze the effects of global voluntary certification in various institutional contexts.

Our initial assumption was that successfully implemented global standards lead to changes in practices at the local level, aiming to contribute to the sustainability of fisheries. At the same time, the features of the implementation of standards are determined by the presence of an international infrastructure, the involvement of local actors in global networks, and the peculiarities of the local context. Therefore, we looked at the global/local interplay of actors in governance using a set of qualitative methods. Semi-structured interviews and document analysis were the main methods used in both cases. Interviews were performed with representatives of fishing companies, NGO experts, and MSC representatives (see Appendix A). Informants were selected using the snowball method. MSC expert reports were analyzed. In total 20 interviews were conducted in 2021–2022: 11 interviews were collected for the Finnish case study, and 9 interviews were collected for the Russian case study. All interviews were transcribed and analyzed by thematic and axial coding [49]. We had the same interview guide for both cases, but it was adapted to national circumstances and also according to the role of the interviewee (NGO or fisheries representative, scientist, business representative) (see Appendix B for the interview guide). The main questions that the informants were asked during the interview were: What actors were involved in the implementation of the MSC? How have fishing practices changed since the introduction of MSC? What difficulties arose during the certification process? How was

the audit conducted? We used the thematic and axial coding method [49]. First, within the framework of thematic coding, we identified the main topics of the interview: the main stages and difficulties in the implementation of certification; the attitude of state authorities to certification; interaction between fishermen and experts; previous experience in marine resource management; international organizations; changes in practices; standards and structure MSC; features of the audit; certification costs; market drivers of certification; the attitude of state actors to certification, etc. Further, on the basis of the identified topics, axial categories were identified, which were analyzed in more detail and connected with theoretical provisions: configurations of global actors involved in the implementation of the MSC; gaps between global standards and local practices; adaptive strategies of local actors; deformation standards. The data was triangulated by comparing information obtained from different sources (interviews with different groups of actors, MSC reports). This allowed us to reconstruct and analyze the specific features of the institutionalization of MSC in the two countries.

## 4. Results

### 4.1. MSC in Finland

#### 4.1.1. GGNs Involved in Fisheries Regulation in Finland (Baltic Sea)

The EU has exclusive competence in the fisheries policy among its member states and plays a significant role in marine governance. Its policy instruments represent a set of specific GGNs. As Finland is an EU member state, Finnish marine fisheries must comply with the Common Fisheries Policy (CFP) and all legislation of the EU. This means that via extensive forums of negotiation, CEP requirements were approved and channeled to all European member states, which represent sites of implementation for CEP. CFP aims at the sustainability of fish stocks and economic and social sustainability of commercial fishing. National fishing quotas, annual total allowable catch (TAC), and extensive control of commercial fishing are the main aspects of EU policy impacting Finnish marine fisheries [53]. Behind EU decisions on annual TAC for different fish species, there is the International Council for the Exploration of the Sea (ICES) scientific advice on fishing opportunities, which is varyingly followed in the EU decision-making on TACs [53]. Therefore, Finnish fisheries have been regulated for years by international (ICES-GGN) and, since the mid-1990s, by regional (EU) regulators. Therefore, several GGNs served as facilitators-promoters for EU fisheries regulation. In addition, there is national Finnish fishing legislation.

The above-mentioned ICES-GGN is an intergovernmental organization that oversees the development of the world's fishing industry and develops standards for the sustainable extraction of marine resources. Its site of implementation is much wider; it was created in 1902 and focused on the study of marine ecosystems in the North Atlantic. Finland and Russia have been ICES members since the beginning. ICES-GGN operates through regular and ad hoc scientific forums of negotiation, for example, working groups that study, negotiate, and recommend fishing quotas. The functions of ICES include conducting scientific research, for example, observing ecosystems and overfishing, as well as developing recommendations for sustainable fisheries. ICES has a large database of the state of marine resources and provides advice on fishing quotas. The ICES Council of Delegates, the node of design of ICES-GGN, has voted to place a temporary suspension on all Russian Federation delegates, members, and experts from participation in ICES activities due to the current geopolitical situation [54].

The Baltic Marine Environment Protection Commission, also known as the Helsinki Commission (HELCOM-GGN), brings the Baltic Sea states, Finland and Russia included, into the same forum of negotiation. HELCOM was established by the 1992 Convention on the Protection of the Marine Environment of the Baltic Sea Area (Helsinki Convention). HELCOM has emphasized minimizing eutrophication—the major environmental threat to the Baltic Sea—but also, for example, ecosystem and species protection has gained attention and a HELCOM marine protection area network has been created. HELCOM objectives and rules are in line with EU environmental law and policies and complement

and support their implementation [55]. The HELCOM Baltic Sea Action Plan also includes fishing-related actions, such as reducing the negative impacts of fishing activities on the marine ecosystem and, to this end, supporting the development of fisheries management including technical measures to minimize unwanted bycatch of fish, birds, and marine mammals and achieving the close to zero target for bycatch rates of relevant species by 2024, especially the Baltic proper population of harbor porpoise by 2022. The Action Plan does recognize bottom trawling as a threat to seabed habitats but does not propose any direct action to forbid or minimize the use of it [52]. Therefore, HELCOM-GGN also serves a facilitating role for MSC-GGN.

### 4.1.2. MSC Implementation in Finland

Baltic herring and Baltic sprat (hereinafter herring and sprat) have been MSC-certified in Finland since 2018, and therefore their habitats became sites of implementation for the MSC-GGN. It is a group certificate managed by the Finnish Fishermen's Association (SAKL). Herring and sprat cover about 90% of the Finnish marine fish catch (Interviews 2021) and are the only MSC certified fish species in Finland.

Already at the beginning of the century there was a preliminary survey about the feasibility of MSC for Finnish fisheries, but the issue did not make progress until 2015 when MSC's Finnish project manager actively brought up the issue and "*also among us, in our executive committee [of SAKL] we pondered about it, and we saw that the Finnish central wholesale business had started to bring this up more and more often ( . . . ) in summer meeting 2015 the committee agreed to find out where to get funding for a feasibility study on certifying Finnish herring fishing*" (Stakeholder interview 2021). SAKL received funding from a Finnish research fund, and the feasibility study was performed in 2016, showing that certification was possible. After that SAKL sought out more funding in order to start the certifying process and received it from the Maritime and Fisheries Fund of the EU. Otherwise SAKL would not have had enough money to start the process (Stakeholder interview 2021). Thus, governmental support for the certification process was essential. Getting an independent certification for the Finnish fisheries has been in line with the policy of the Finnish bioeconomy [56].

WWF-GGN was not active in initiating MSC in Finland, but has been cooperating with MSC in, for example, raising awareness of sustainable fishing by joint campaigns and events (Stakeholder interview 2021). However, the indirect effect of WWF has been bigger because the role of the WWF seafood guides is considerable for Finnish consumers and especially for professionals, retailers, etc. Earlier WWF has internationally always considered MSC-certified fish as 'green', that is, recommendable, but international WWF has made a decision that WWF no longer automatically gives the green light to MSC-certified fish, which is considered a problem in Finland by other stakeholders: "*it gives such a signal that this MSC certificate is not reliable in general, so it is a signal that undermines the value of this certificate*" (Stakeholder interview 2021).

In 2019 a partial self-suspension of the certificate of the Baltic Sea main-basin herring took place due to a negative ICES assessment [57] and a statement of the auditing company. The herring population of the Gulf of Bothnia (the Northern part of the Baltic Sea) is still sustainable, and these fisheries are still MSC-certified.

There has been discussion about also certifying other species, but either the fish catches are so small that certifying is not feasible or the surveillance of scattered fish stocks in lake areas is currently practically impossible. "*So if we are realists, only herring and sprat are economically relevant species, at least within the current MSC system*" (Stakeholder interview 2021).

The paperwork of MSC is mostly concluded by SAKL: "No, not really, as I answered earlier, since we have been following these [EU rules] earlier, so effects are rather small. We must do certain reports, which were not done earlier, to the auditing company, but for fishing companies nothing new has come, since we do this work for them, as this a group certificate" (Stakeholder interview 2021). However, the fishermen have been facing the tightening surveillance by the EU, as described by an interviewee: "yes, mainly it is

this kind of micromanaging. Fishing has always been considered as a free occupation, but it is not anymore today. Every step is being watched and in the future every fisherman at the sea will be controlled electronically, so the 'big brother' mechanism is on. It is not considered good" (Stakeholder interview 2021).

### 4.1.3. MSC and Sustainable Fisheries in Finland

The Baltic Sea is a complicated site of implementation for MSC-GGN. Even if eutrophication and climate change are the main threats to the marine environment in the Baltic Sea, fishing also has several negative effects. There are direct effects on target species, and bottom trawling and bycatch affect many other species and habitats as well. Fishing also contributes to shifts in the food web and size-age distribution and reduces the resilience of both fish and other marine organisms [55]. Finnish fisheries did not include bottom-trawling even before the MSC certification, so the effect of MSC on actual fishing methods of the Finnish fisheries has been small. Yet, MSC has made fishermen place more emphasis on the surveillance of fishing, especially related to bycatch. "*It [MSC certification] has many effects, increasing general awareness of sustainable fishing. Requirements of certification are also relatively tight. In Finland or even in the Baltic Sea the problem is not evaluation of fish stocks and keeping it on the MSY level, but as for Baltic herring and sprat, we worked mostly on bycatch, which gets less attention in ICES advices. ( . . . ) This has not yet affected fishing tools, but it has changed things so that more information is being collected on bycatch and let's see whether there will later be need to change fishing, too*" (Expert interview 2021).

According to several interviewees, MSC has also strengthened the role of scientific knowledge in political decision-making. It is now less likely that stakeholders and politicians would speak against ICES advice on sustainable fishing levels in favor of short-term economic interests (Expert interviews 2021) [58].

MSC has not offered price premiums in Finland. Rather, in Finland, MSC has first and foremost guaranteed access to internal markets, as Finnish central wholesale business and retailers have started to demand MSC certification. In some cases, international buyers also ask for the MSC certificate, but the main reason to certify is the internal Finnish market. In 2021, almost half of the Finnish export of fish and fish food was targeted to Russia, Belarus, and Ukraine, where the demand of MSC does not yet exist [59].

EU fishing regulation is constantly tightening and might dilute the extra benefit of MSC surveillance. However, the role of MSC-GGN in ensuring access to markets and bringing the scientific knowledge on ecological sustainability into multiple forums of negotiation within the EU is not likely to be diluted.

### 4.1.4. Criticism of MSC in Finland

The current certification of herring and sprat was not much criticized in the interviews. Stakeholders consider that they have been able to meaningfully participate and have a real effect on the evaluation process: "*Actually what experience I have in the Baltic Sea region and fisheries there, in all process where we have been involved in a way or another, our observations and remarks have always been well received, so I don't have any complaints, that our opinions would not have been taken into consideration, on the contrary, they have always been taken into consideration, and we have been listened to*" (Stakeholder interview 2021).

Yet, the price and paperwork of the MSC certification has been considered burdensome by some actors. Even if MSC is often considered very bureaucratic, EU requirements for fishing received even more criticism from the interviewee(s).

Finnish small-scale fishing is so scattered and happens in part in the thousands of lakes of Finland that fishermen cannot afford MSC certification, and it is also practically impossible to study the state of fish stocks in every single lake. A need for an "artisan fishing certificate" was acknowledged (Interviews 2021). An interviewee worried about the need for certification spreading even to this kind of small-scale fishing said: "*There is nothing wrong with MSC, how it works etc., but we worry that if all fish should be MSC certified,*

*and it is not possible for all fisheries, and as we have spoken today, it (certification) does not even bring any added value, so this is perhaps our concern"* (Stakeholder interview 2021).

*4.2. MSC in Russia*

4.2.1. GGNs and National Regulatory Agencies Involved in Fisheries Regulation in Russia (Barents Sea)

In Russia, the governance of marine resources is carried out by the Federal Agency for Fisheries (Rosrybolovstvo), subordinate to the Ministry of Agriculture of the Russian Federation. The federal agency has territorial subdivisions that regulate the fishing industry at the regional level. The Russian marine biological resources management system includes: monitoring, protection, and reproduction of fish resources; control over compliance with legislation, development, and implementation of measures to support fisheries; study and conservation of biological resources and their habitats; and organization of fish processing [60]. In 2020, the maritime doctrine of the Russian Federation was approved, according to which the goals of the maritime policy of the Russian Federation are: the implementation and protection of domestic fisheries in the world ocean; development of principles for sustainable management of marine resources; and conducting scientific research and developing a monitoring system for marine ecosystems.

However, the management of marine fisheries is traditionally very globalized, which is explained by the specificity of the resource and the inability to maintain sustainable fisheries within the boundaries of a single country [17,61].

In this regard, international arrangements play an important role in the regulation of Russian marine fisheries. In the case of the Murmansk region and Barents Sea, several GGNs play an important role.

In Russia, as in Finland as well as the rest of the world, one of the key players is ICES-GGN, the operation of which we described in the previous sections of this paper. Another important international actor is the North East Atlantic Fisheries Commission (NEAFC), which regulates marine fishing in the Atlantic and Arctic Oceans. It represents another facilitating GGN. NEAFC-GGN makes fishery regulatory decisions based on scientific advice from ICES and other scientific research. Within NEAFC, a number of special scientific groups have been organized as a forum of negotiation to analyze certain issues of fishery regulation [50]. Another GGN, the Joint Norwegian-Russian Fisheries Commission (JNRC-GGN) is an important actor in the Barents region. The regulation of fisheries in the Barents Sea is developed within its node of design. JNRC monitors and organizes joint scientific research, as well as sets quotas for sustainable fisheries in its Barents Sea site of implementation [61]. In the same way as in the Baltic Sea, the Barents Sea operated several GGNs for many years. Since ICES and JNRC have been translating international requirements for fishing in the Barents region for many years as well, the arrival of MSC certification in the Murmansk region is not something completely new for the fishing sector. Moreover, many of the requirements of the MSC certification were based on international agreements and coincided with the requirements for sustainable fisheries which were presented by ICES-GGN and JNRC-GGN [62]. Therefore, NEAFC-GGN, ICES-GGN, and JNRC-GGN served as facilitators-promoters of MSC-GGN.

Therefore, since the requirements for marine certification are based on international agreements and conventions and coincide with the requirements for sustainable fisheries that were made by ICES-GGN and JNRC-GGN, it paved the way for the arrival of the MSC-GGN.

4.2.2. MSC Implementation in Russia: Barents Sea Case Study

One of the first Russian regions where maritime certification began to develop was the Murmansk region [63]. In 2010, the first certificates for cod and haddock fisheries were obtained. The demand for marine certification in the region has been driven by a well-developed fishing industry and its focus on international markets. *"Certification is screwed out because a lot is now being exported abroad, and companies don't take anything from*

*you without a certificate even for a lower price.*" Or: "*I believe that there is nothing wrong with that, but the main driving force for any commercial structure is money and profit. If money and profit help us to preserve nature, then why not to certify*" (Stakeholder interview 2021).

In 2021, companies in the Barents Sea received certificates for the fishery of cod, haddock, shrimp, pollock, halibut, opilio, and Kamchatka crabs [64].

A distinctive feature of MSC-GGN in the Barents Sea is that the major certificate holder is the Association of Fishing Industry of the North, which unites a large number of fishing companies in the Murmansk region. This made the certification process much easier and cheaper for individual companies. However, the largest fishing companies in the Murmansk region have both group certificates as well as their own. We have identified several main aspects of the interaction of global and local actors during the institutionalization of MSC.

The institutionalization of global rules in Russia, even in the case they are private regulatory schemes, presupposes their acceptance by state actors. Initially, the Russian authorities reacted to the new system with distrust and expressed ideas for the creation of a national certification system. However, it became clear that the state-based administrative level of control would not allow the creation of a certification scheme that is in demand on world markets. It must be based on market mechanisms and be recognized by other national and global actors. "*Our officials have been saying all the time, let's create Russian certification. And then they came to some Marubeni company with their own certificate, and the impolite Japanese said, hang this certificate on a carnation in the toilet, we will not recognize*'" (Stakeholder interview 2021). The need for MSC certification for international trade has contributed to the emergence of a more loyal attitude of the authorities towards certification. During the certification process, MSC auditors regularly meet with representatives of the Territorial Marine Resources Administration and discuss existing problems. At the same time, the authorities are still wary of maritime certification. Representatives of MSC certification bodies are not viewed by the Russian authorities as legitimate or influential players whose opinion should be heeded. Therefore, their recommendations are not always taken into account by the authorities when developing any government programs or strategies, unlike, for example, the recommendations of such intergovernmental organizations as the JNRC or ICES. "*Well, whatever they recommend to me, I understand that we could not move our Russian colossus with these recommendations*" (Stakeholder interview 2021). In addition, the authorities are reluctant to provide auditors with all the necessary information, citing the fact that fish is a strategic resource. It also demonstrates a lack of confidence in the system on the part of the Russian authorities. "*I actually tried to find out why it is so difficult to get access to information on catches, for example; officials say to me: fisheries are a strategic resource . . . do not want to give the information away. Almost a military secret turns out*" (Expert interview 2021). Thus, the Russian authorities do not hinder nor help the introduction of MSC certification in Russia.

In autumn 2022, half a year after the beginning of the Russia–Ukraine military conflict, MSC certificates in the Barents Sea were active, audits were taking place as planned, and auditing firms were overcoming difficulties in bank transfers (Stakeholder interview 2022).

### 4.2.3. MSC-GGN and Sustainable Fisheries in Russia

There are several main effects of MSC-GGN in governance of fisheries, markets, and on fishing practices at the site of implementation, the Barents Sea. First, it is the stabilization of the position of Russian fishing companies in international markets. As the managers of the companies note, the MSC certificate allows them to work with international partners. In addition, it makes it possible to supply fish to the market at a higher price. According to an expert estimate, the price of certified products in international markets is on average 10–15% higher. At the same time, MSC-certified fish on the domestic Russian markets is still not in demand. The second important consequence of MSC certification is the significant greening of the fish industry. Thus, in the Murmansk region, bottom trawling had a serious impact on the marine ecosystem [53]. MSC requirements have spurred scientific research into more sustainable fishing techniques to mitigate these negative impacts. In addition, as

part of the certification, research was carried out to identify untouched areas of the Barents Sea (about 14 thousand miles), which were excluded from the routes of fishing vessels. This will help preserve valuable marine ecosystems. In this regard, marine certification has brought new incentives and practices over the requirements of JNRC-GGN and ICES-GGN, which are focused on stock assessment.

Therefore, MSC certification has contributed not only to governance arrangements, but to an increased focus on impacts on marine ecosystems and the development of measures to promote their conservation. Third, MSC-GGN contributed to democratic decision-making: it fostered cross-sectoral dialogue between companies, governmental agencies, and NGOs. Certification requires multiple forums of negotiations with stakeholders, including public authorities, and NGOs during an audit. In addition, the certification created opportunities for NGOs to influence companies and pushed businesses to cooperate with experts. For example, MSC's requirements for greening fishing have encouraged cooperation between WWF and fishing companies. *"This is the only lever? In which WWF is dealing with the fishermen. If there were no MSC, then we would not have any dialogue"* (Stakeholder interview 2021). Finally, MSC certification has increased the role of scientific expertise in marine resource extraction. Assessment of the sustainability of the fisheries, as well as the fulfillment of MSC standard, requires scientific research data. In particular, in the Murmansk region, such expertise is provided by the Polar Branch of the All-Russian Research Institute of Fisheries.

In general, experts and managers of fishing companies note a gradual evolution in the behavior of fishing companies compared to the Soviet and post perestroika periods, which is due to the emergence of a longer-term horizon for planning fishing activities, the connection of market advantages with more environmentally friendly fishing practices, increased transparency, and increased interaction with various stakeholders. *"Companies are already sitting and thinking about how they will work in 3–4 years. They paid billions for these new ships, and they do not need these ships, which can catch 30–40 thousand tons per year, to stand idle. Therefore, it is better that they hold themselves a little this year in order to see the perspective for 3, 4, 5 years. Therefore, certain evolution in the minds of fishermen is also taking place"* (Stakeholder interview 2021).

### 4.2.4. Criticism of MSC in Russia

In Russia, MSC-GGN is a vulnerable arrangement. Market In Russia, MSC-GGN is a vulnerable arrangement. Market actors always desire to reduce their costs. This can lead to the weakening of the MSC standards in the process of their implementation. Local turbulent context can also affect the implementation of MSC standards. The MSC standard requires effective monitoring, robust data collection, and surveillance to be in place. Therefore, observers are usually deployed in certified fisheries and additional controlling observers are placed onboard the vessels.

In Russia companies tend to exaggerate the number of observers and include border guards performing completely different functions as observers. In some cases, observers are people who do not have the necessary knowledge and, accordingly, are not able to carry out quality control. Sometimes the fishermen try to restrict the observation opportunities on ships. *"They counted all the border guards who were on the ships and included them as the observers. And when you ask: what are these border guards watching, how they transmit data . . . Well, in general, the observer system itself needed to be improved, because people were often untrained"* (Stakeholder interview 2021).

For the sustainability of fisheries, the preservation of young fish stock is important. This means that young fish that have been caught should be released back into the sea. However, in some cases, fishing gear is used to throw young fish overboard, which damages the fish: *"In particular, I raised the question that fish is thrown away from the deck . . . Imagine how the fish will survive after that"* (Stakeholder interview 2021). In addition, some MSC standards are not relevant for Russia, so monitoring their compliance is not relevant. For example, in recent years, the MSC scheme has begun to pay attention to the social aspects

of fishing, which is reflected in the emergence of new requirements, for example, to have no child or slave labor, which is not relevant for Russian fisheries.

Thus, during the implementation of the MSC standards, some variation in them is possible, which is explained by the impossibility of controlling all practices as well as by the peculiarities of the local context.

## 5. Discussion

We applied the GGN theory in order to conceptualize our findings and understand the global/local interplay of actors in governance of fisheries. GGN represents the phenomenological explanatory concept, which is appropriate for institutional analysis and explains how global standards developed transnationally are translated, adapted, and implemented locally by global to local networks operating across different scales [23,65–68].

We determined several GGNs, which serve as facilitators/promoters of MSC-GGN institutionalization in both Finland and Russia. Most of these GGNs are driven by international agreements and conventions, but there are also private GGNs, such as the WWF and ISEAL alliances. Some regulatory facilitating GGNs are global, while others are regional. ICES represents a global regulatory GGN, while the EU Fisheries Policy facilitates MSC implementation in all Baltic Countries (except for Russia), for instance in Estonia, Sweden, and Finland. The Russian-Norwegian Cooperation Agreement tackles Russia and Norway only in the Barents Sea, and NEAFC-GGN affects two oceans, that is, the Arctic and Atlantic. Sites of implementation in MSC-GGN in both Finland and Russia turned out to be shared with larger regulatory GGNs. This makes it easier for companies to comply with MSC in both Finland and Russia as they already have to comply with requirements of other, larger regulatory bodies. On one hand, this facilitated companies entering MSC, and on the other hand, the added value became less obvious.

Therefore, the institutionalization of marine certification in both countries was facilitated by the tight international regulation in this area involving international agreements. An important role was played by such GGNs as ICES, NEAFC, and JNRC, which conducted scientific research via multiple forums of negotiation, established the sustainable annual catch, and offered ways to preserve marine ecosystems. As for Finland, the EU had already established stringent control of fishing, so MSC-GGN did not bring much new work for fishermen. The substantial role of global GGNs in the study areas made it possible to avoid institutional confrontation between the MSC standards and the existing fishing rules in these countries.

Certification GGNs are market-driven regulatory tools. However, the institutionalization of MSC in Russia and Finland had a number of similar and distinctive characteristics (see Table 1). In both cases, the introduction of MSC was connected to the interest of national companies on the international markets. This became a major incentive for companies especially in Russia to get certified. In Finland the requirements of Finnish central wholesale business were as strong or an even stronger incentive. In addition, there were internal sensitive markets in the European Union, which increased the market motivation of Finnish companies to certify their products. At the same time, most Finnish fishing companies were small, which made MSC certification expensive for them. In Russia, there were no domestic sensitive markets that would provide domestic demand for certified goods, so certification was spreading in those regions that were connected with the international markets (e.g., in the Murmansk region). Despite the large number of small fishing companies in the Murmansk region, they were able to obtain the MSC certificate thanks to the activities of the Association of Fisheries of the Murmansk region, which organized the obtaining of the group MSC certificate.

In both Finland and Russia, most of the social standards required by MSC, such as no child labor, which is an important issue in developing countries [69], were not relevant. This made it easier to implement MSC.

**Table 1.** The main features of MSC implementation in Finland and Russia.

| The Main Features | Finland | Russia |
|---|---|---|
| Incentives for MSC | International and domestic market requirements | International market requirements |
| Regulatory drivers of MSC | ICES, EU regulation | ICES, JNRC, NEAFC |
| Role of WWF | Reducing of MSC support, but participation in MSC events | Active role in the implementation of MSC |
| Role of the state | Financing, political support | Distancing from MSC |
| Effects of MSC | Improvements of the fish bycatch, strengthening the role of ICES & science | Alternatives to bottom trawling, partnerships between WWF experts and fishermen, long-term planning |

The implementation of the MSC in Russia and Finland required the efforts of international and local actors to adapt the new rules to the local context. Partnerships between fishermen and various expert organizations have been an important adaptation strategy to help the companies to meet the new requirements. In Russia, MSC was implemented with the active participation of WWF, which contributed to the development of more environmentally friendly fishing practices. In Finland, the WWF was not so tightly involved in the maritime certification process, however, it participated in some joint information events activities. In both cases MSC implementation required close cooperation with scientific institutions and using scientific data. An important factor in the successful institutionalization of the system was the attitude of state authorities, which could support it or, conversely, hinder it. In Finland, the state had a positive attitude towards MSC and the SAKL received financial help from an EU fund, which reduced the costs of the institutionalization of MSC in Finland. In Russia, the state had a wary attitude towards MSC, which was due to the general level of distrust in Western organizations and non-state actors. At the same time, international rules developed on the basis of intergovernmental agreements were perceived by the Russian state more positively than global standards initiated by non-state actors.

The effects of MSC implementation were associated with the strengthening of already existing international standards for the extraction of marine resources. Its implementation contributed to a more effective implementation of already existing international environmental standards and to a more robust role of scientific knowledge. In both countries MSC drove additional awareness, transparency, and strengthened good fisheries management. According to the informants, in Finland, MSC has not brought fundamentally new fishing techniques. This was partly due to the fact that the fishing methods most detrimental to marine ecosystems (e.g., bottom trawling) were not used in Finland even prior to MSC certification. The most significant requirements for the Finnish fishing industry were the MSC requirements for fish bycatch. In Russia, experts noted more significant environmental and social effects from MSC. In the Murmansk region, the introduction of MSC stimulated closer cooperation between WWF experts, fishermen, and research institutes. Thus, MSC-GGN has contributed to the establishment of an ongoing forum of negotiations between the WWF and the fishermen. Within the framework of this cooperation, gentler fishing techniques were developed and implemented. In particular, an alternative to the bottom trawl was found. Informants also noted the gradual formation of ideas among Russian fishermen about sustainable fishing and the need for long-term planning. However, as experts noted, part of the environmental standards were transformed during the implementation. For example, this concerned the requirement for observers to be present on-board fishing vessels or the need to release excess bycatch.

Sometimes the international actors can also rock the boat they have created. In Finland that happened when WWF International withdrew the automatic support for MSC certified fisheries in their Seafood Guides. In Finland this was considered negative, but it can also help to maintain the integrity of sustainable seafood business by bringing up

possible defects. As we can see, WWF-GGN played a significant role in facilitating the institutionalization of MSC-GGN, while in Finland this was not the case.

Thus, the institutionalization of global standards at the national level was determined by the global nodes of design of already established GGNs, with multiple stakeholder forums of negotiation in each GGN and between GGNs. The density of international GGNs in both the Baltic and Barents Sea, the peculiarities of the national management system, the availability of serious scientific and expert support, and dependence on international markets turned out to be important.

## 6. Conclusions

Our paper demonstrates that both the Baltic Sea and the Barents Sea fish stocks are evaluated, monitored, and regulated by a significant number of public and private regulatory bodies. Multiple GGNs act along with voluntary MSC-GGN towards the sustainability of marine resources at the sites of implementation. These GGNs involve both public and private actors. Involving stakeholders, such as NGOs, fishing cooperatives, and scientists contributes to democratic decision-making in fisheries management. The market is an important driver for sustainable fish production, but it is not the only one, and the MSC label is not the crux—rather it is part of a broader set of benefits (and costs) associated with MSC, which jointly deliver a comprehensive value proposition to the users of the MSC program. Therefore, all GGNs involved have a cumulative effect on sustainability of particular fish stocks. The paper demonstrates that MSC contributes, albeit in a limited way, to both the economic stability of certified companies and biodiversity conservation. The administrative work, however, is a burden for fishing companies in both Finland and Russia as they need to report to several regulatory bodies, involve scientific institutions, and pay consultants.

An analysis of the interaction between global and local actors in the process of introducing certification in two countries made it possible to identify several important futures. More added value of MSC certification turned out to be in Russia rather than in Finland despite the context for the introduction of MSC being more favorable in Finland. In Finland, the state authorities supported the certification, while in Russia the state reacted with disbelief. Finnish companies were ready to disclose information about their activities and followed the principles of transparency required by certification. Certain ecological fishing practices existed in Finland already before MSC. For example, there were no problems with the bottom trawling that existed in Russia. This has led to less added value of MSC in Finland. Despite the greater number of difficulties in Russia in MSC implementation (distrust of the state, lack of transparency, less environmentally friendly fishing practices), the added value of MSC turned out to be more significant. Therefore, MSC-GGN closed the gaps in Russia's democratic decision-making processes regarding management of marine resources and addressed gaps in regulation.

MSC-GGN has peculiarities in transferring the standards developed in the nodes of design to the sites of implementation. On one hand, the dissemination of global standards is aimed at spreading similar practices of environmental and social responsibility to companies around the world. On the other hand, the successful institutionalization of global standards at the local level is possible only if they are flexible and vague enough. MSC gives the general direction for development, while specific practices are developed during forums of negotiation. The possibility of different filling/interpretation of global standards at the local level makes these standards more attractive to local actors. Yes, practices of certified companies turned to be different in different countries, as well as in Finland and Russia

Our findings demonstrate the important role of facilitating GGNs; the layering of rules and regulatory structures can lead to increased regulatory outcomes when several GGNs are operating on a single site of implementation. Thus, the positive effects of the MSC-GGN were partly due to the fact that other GGNs (ICES, JNRC, NEAFC) had already created a

favorable institutional environment in the particular sites of implementation in both Baltic and Barents Sea.

It is important to note that there are not only top-down effects in GGN operation, but also bottom-up. The operation of MSC-GGN has strengthened the role of ICES-GGN in the Baltic Sea and the EU. Therefore, existing gaps or failures in governance can be corrected through the cooperative capacity of several GGNs, which result in additional regulation as well as better implementation of global standards.

## 7. Perspective for Future Research

The research team is interested in studying the dynamics of private regulation related to the current geopolitical situation in the world. The private certification and labeling schemes are constantly changing and remain fragile regulatory tools. In Finland, MSC is small, the Program of Endorsement of Forest Certification (PEFC) is widely spread, and FSC has been expanding in the 2020s. In Russia, MSC in the Barents Sea remains stable despite the Russian attack on Ukraine in February 2022. FSC chain of custody certificates are suspended by FSC-International and the EU's 5th package of sanctions, while FSC forest management is still in operation. PEFC is suspended. New national forest certification schemes, such as Forest Etalon (substitute of FSC) are emerging in Russia. Suspensions can cause logging of virgin forests and violations of indigenous people's rights in Russia. The 5th package of sanctions does not allow the trade of wood between Russia and Europe; however, fish exports are not under sanctions currently. The situation in the markets of natural resources is constantly changing, and newly established markets, involving non-certified companies, may be less environmentally 'sensitive' than, for example, European markets. The development and sometimes the occasional decline of certification and labeling schemes need to be studied in order to understand the emerging solutions for and effects on resource management during changing geopolitical situations.

**Author Contributions:** Conceptualization, S.T., M.T. (Maria Tysiachniouk), M.P.; methodology, M.T. (Maria Tysiachniouk), S.T.; formal analysis, S.T., Tysiachniouk, M.P.; investigation, S.T., Tysiachniouk, M.P., M.T. (Minni Tynkkynen); writing—original draft preparation, S.T., Tysiachniouk, M.P., M.T. (Minni Tynkkynen); writing—review and editing, S.T., Tysiachniouk, M.P.; visualization, M.T. (Maria Tysiachniouk); Project administration, M.P.; Funding acquisition, M.P., M.T. (Maria Tysiachniouk), and S.T. All authors have read and agreed to the published version of the manuscript.

**Funding:** This research is part of the project 'Confronting sustainability: governing forests and fisheries in the Arctic' (ConSust), funded by the Academy of Finland, grant number 346655.

**Institutional Review Board Statement:** Not applicable.

**Informed Consent Statement:** Informed consent was obtained from all subjects involved in the study.

**Data Availability Statement:** Not applicable.

**Acknowledgments:** We thank the anonymous referees and all the interviewees for their invaluable input. We thank Alexandra Orlova and Sofia Beloshitskaya for illustrating the article.

**Conflicts of Interest:** The authors declare no conflict of interest, and the funders had no role in the design of the study; in the collection, analyses, or interpretation of data; in the writing of the manuscript; or in the decision to publish the results.

## Appendix A

Stakeholder interviews in Finland: Representatives of KKL (the Federation of Finnish Fisheries Association) (two interviewees), Kotipizza Group (one), MSC Europe (one), MSC Finland (one), SAKL (Finnish Fishermen's Association) (one), and WWF Finland (one). Expert interviews in Finland: Science representatives of the University of Jyväskylä, Finland (two interviewees) and Luonnonvarakeskus (Finnish Natural Resources Institute) (two).

Stakeholder interviews in Russia: Representative of Regional Association of Fishermen (one), MSC certification body Russia (two), WWF Russia (one), WWF Murmansk (one). Expert interviews in Russia: Polar Research Institute (one), Saint Petersburg State University (one), fishing companies (two).

**Appendix B**

Interview guide

1. What are the main incentives for the implementation of maritime certification?
2. What actors were actively involved in the implementation of the MSC?
3. What was the role of the WWF in MSC implementation?
4. What obstacles arose in the process of MSC implementation?
5. What was the role of government actors in the process of MSC implementation?
6. What was the role of scientific organizations during the MSC implementation?
7. What international regulation was in place in these regions prior to maritime certification?
8. What factors influence the specifics of the MSC institutionalization in each country?
9. How do global standards adapt to the local context within MSC?
10. How was the audit conducted?
11. What fishing practices have been changed by marine certification?
12. What are the main environmental implications of marine certification?
13. What are the main social implications of marine certification?
14. What strategies were used by companies in the process of MSC implementation?
15. What deformations occurred with the global standards in the process of their implementation?
16. How has the interaction between business, NGOs, state authorities, scientists been changed in the process of MSC implementation?
17. What is the added value of MSC in fostering sustainable fishing in Finland/Russia?

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
