# Peer review of "Marine Stewardship Council Certification in Finland and Russia: Global Standards and Local Practices"

_sustainability, doi:10.3390/su15054063_

Round 1

Reviewer 1 Report (Previous Reviewer 4)

The methodology for selecting indexes and quantifying them and evaluation in a logical frame work is important. Statistical analysis for validation must describe. you must quantify some indicator and index to evaluate the condition.

The index and indicators for assessing the sustainability, ecological services of sea and water quality status must describe.

The index and indicators for assessing the sustainability, ecological services of sea and water quality status must describe.

The logical framework for analysis of driving force and state and response interaction is necessary. The following works are good sample for systematic analysis: Integrated groundwater management using a comprehensive conceptual framework, Flood assessment in the context of sustainable development using the DPSIR framework. You can also improve literature by them.

Important note about social indexes and their relation with sustainable fishery in the area must discussed in conclusion. It is important that main contribution and results of the paper presented and discuss in the abstract.

There are many indexes which quantifying them are hard and important. For example how do you quantify indexes such as: willingness of stakeholders for collaboration in sustainable fishery? Or how do you quantify Environmental awareness or how do you consider social learning of fisher and their knowledge sharing about sustainability.

More figures and chart is necessary for generating the results and discussing about relation between indexes and fisher reactions .

Conclusion section is too short. In conclusion section, you should discuss about importance of your work and the works must do to improve the sustainability. At conclusion you must reply to the question which is title of the paper.

Author Response

Dear Reviewer,

We are pleased to resubmit our revised paper “Marine Stewardship Council Certification in Finland and Russia: Global Standards and Local Practices”. We are very grateful for the comments and advice. Your thoughts are inspiring, but we would like to keep our phenomenological approach in our paper and use qualitative methodology. We made our general phenomenological approach much clearer, however we keep the main aim of the paper as originally designed. We significantly revised our introduction, methodology, results, discussion and conclusions.  The paper became very different.  Now our paper is better referenced.

Reviewer 2 Report (Previous Reviewer 2)

This Manuscript has been improved a lot. However, there are still some points require improvements. Some of my comments are list as below,

Figure 1 was missed.

The spaces before the period are strange, like line 43, 47, 49, 51 etc. Please check it carefully.

Some Abbreviations are still not given full names, like MSI in line 170, and the full name should be given just at the first time the concept appeared, like ICES in line 495.

Line 354, the link show be moved to the reference.

Line 559, I recommend to use a neutral word instead of invasion.

Line 770, Is the content in brackets based?

Also, I still suggest to use the tables to improve the comparison between Finland and Russia. Are the interview questions the same between Finland and Russia? The original questions and interview answers can also be supplied in the format as supplementary materials, if possible, with the more detail information of the interviewees.

Author Response

Dear Reviewer,

We are pleased to resubmit our revised paper “Marine Stewardship Council Certification in Finland and Russia: Global Standards and Local Practices”. Thank you very much for your important and helpful suggestions.  We very much appreciate that you thoroughly read and analyzed our paper and we were happy to do improvements that you suggested. We improved all sections of our paper and enriched the reference list.

Reviewer 3 Report (Previous Reviewer 3)

This paper had been conducted an investigation of the seafood resources based on the Marine Steward Council (MSC) along with the Governance Generating Network Theory. Two cases had been studied into the Baltic sea (in Finland) and the Barent sea (in Russia) for this research. This study is very significant to maintain the ocean resource as well as its species. However, there are some drawbacks that the authors should address them to improve their research.

  1. In the abstract, the contribution of this study is inadequate for a publication indexed in Web of Science (WoS) like Sustainability Journal.
  2. There are a lot of typing mistakes which have been found in this paper.
  3. In part 2, kindly re-write this sentence accurately: “The Marine Stewardship Council (MSC), was launched to help tackle overfishing and to drive improvement in fisheries management with an attemptwith attempt to solve the tragedy of the sea commons.”
  4. Additionally, part 2, the background of Marine Stewardship Council is not neccesary to reveal for the international research paper.
  5. The contribution of this research is not outstanding and inadequate both the proposed methodologies and its scientifics.

Author Response

Dear Reviewer,

We are pleased to resubmit our revised paper “Marine Stewardship Council Certification in Finland and Russia: Global Standards and Local Practices”. Thank you very much for your important and helpful suggestions.  We very much appreciate that you thoroughly read and analyzed our paper.  You mentioned that our study is very significant for maintaining ocean resources and its species.  Maybe indirectly. Our aim, however, is to explain global governance of natural resources on the example of MSC in interaction with other global/local arrangements and the interplay of actors in policy implementation.  Our paper is a social science paper with emphasis on institutional analysis and its innovation is in highlighting peculiarities of global marine governance arrangements. We significantly revised all sections of the paper, and enriched our reference list.

Round 2

Reviewer 1 Report (Previous Reviewer 4)

The paper is well revised.

Author Response

We are very grateful for the comments and advice. Your thoughts were inspiring. We significantly revised our paper under your suggestions. Now our paper is better structured and consistent.

Reviewer 3 Report (Previous Reviewer 3)

The authors had addressed the comments from this reviewer. However, there are some weakness points that the authors should address them to improve this study.

  1. There are a lot of typing mistakes which have been found in this revised paper.
  2. In part 2,  Background: the Marine Stewardship Council (MSC), what is the purpose of this part? In  the research paper, the literature review or recent research publications should be provided and analyzed clearly to outline the objects of study. However, this revised paper has not still done it intensively.
  3. The quantitative results must be provided to make clear the MSC method to two cases in Finland and Russia based on the questionnaire investigation.

Author Response

Dear Reviewer!

Thank you very much for your important and helpful suggestions.  We very much appreciate that you thoroughly read and analyzed our paper and we were happy to do improvements that you suggested. We deleted the section “Background of MSC” and expanded the fragment with literature review. We tried to rewrite this part more analytically and indicated the main areas of MSC research. It helped us to describe the specific of our approach more clearly. However, we keep our phenomenological approach and qualitative methodology. We are happy to answer your questions

Comments to the Authors

In part 2, Background: The Marine Stewardship Council (MSC), what is the purpose of this part?

Response

The main goal of this section was to give the brief overview of MSC history and its structure. It could be fruitful for further analysis. In accordance with the reviewer’s recommendations we deleted this section. A small fragment of this section was removed to the Introduction. We hope it makes the structure of the article clearer and more consistent.

Comments to the Authors

In the research paper, the literature review or recent research publications should be provided and analyzed clearly to outline the objects of study. However, this revised paper has not still done it intensively.

Response

We expanded the fragment with literature review and indicated several main directions in the study of MSC. Based on this brief review we emphasized specific of our approach. Also, we included the latest research papers on MSC. For example: Pierucci et al  2022; Murphy at al 2022; Lang, B.; Conroy 2022; Lajus et al 2018; Pappila, M.; Tynkkynen 2022; Keskitalo et al 2022.

"Controversial evaluations and effects of the MSC impact on sustainable fisheries have been reflected in the academic debate [13,14,15,16]. Several main directions of changes under the influence of MSC are identified in the literature: (a) structural effects, such as changes in markets or in power relations [17,18,19,20]; (b) cognitive effects such as construction or promotion of certain discursive frameworks [21]; (c) regulatory effects, such as the influence of transnational rules and standards on government regulation processes, [22,23]. Researchers use different approaches to explain the stability/instability of the MSC certification. Some researchers focus on the opportunities and limitations of market instruments and market competition to increase sustainability in natural resource extraction [17,24,25,]. These papers highlighted the effectiveness of ecolabels, including MSC [17,26,27].  There is also research that looks at stakeholders attitudes and recognition of the label in the supply chain [19]. Another part of the researchers analyzes the embeddedness of MSC in global governance. They show how the layering of different international regulations can strengthen or weaken marine certification [28,29]. Some researchers identified and measured environmental impact [30,31]. Another group of researchers focuses on the interaction of maritime certification with national regulation [32,33]. They demonstrate that the MSC can influence the adoption of stronger national regulation for the development of sustainable fisheries. However, in some cases, MSC may allow government regulation to be “covered” by less stringent rules and standards and used as an excuse for inaction [33].  

Our approach is slightly different. In this paper we will look at MSC as a global governance instrument with its special characteristics and compare institutionalization of MSC in Finland and Russia.  We will look at the interplay of different global regulatory instruments'.

Comment to the Authors

The quantitative results must be provided to make clear the MSC method to two cases in Finland and Russia based on the questionnaire investigation.

Response:

Thank you for pointing out these issues. However, we feel that you are asking us to write another paper with a totally different methodological approach instead of improving the current paper. Below we are doing our best to explain our approach to this phenomenological study.

Our research questions do not imply the quantitative analysis of the level of sustainability or measuring impacts on the state of the environment.  We look at practices that certified companies have to change due to MSC assuming that if these practices are implemented properly the policy tool will work for a better environment. Our main focus is institutional changes under MSC and its wider social implications. We analyzed how MSC standards influence the rules of interactions between companies, NGOs, state authorities, scientific organizations, and local communities. It is really hard to quantify such aspects as environmental awareness or cooperation of different stakeholders. The evidence is expressed in citations from our interviews, highlighting views of MSC staff and stakeholders. We used qualitative methodology in our research. Semi-structured interviews and observations help to reveal main institutional aspects of the MSC as a tool for democratic governance of water resources and highlight its added value.

There should be coherence between the research questions, the theoretical framework, the results and discussion.  We were choosing the theoretical framework that would be the best for a phenomenological study of network governance involving MSC and other regulatory instruments. 

To explain the value of our paper we included at the end of the paper the following paragraph:

This paper contributes to the understanding of global governance with voluntary certification as a tool.  We analyzed different global and national regulatory instruments and highlighted the added value of MSC as a newly introduced regulatory institution.  GGN was widely used by many authors to explain the operation of the Forest Stewardship Council (FSC) and governance of oil.  This paper is the first in using this explanatory framework in Marine governance. The paper looks at  MSC-GGN structure and complex agency.  GGN allows us to look at both agency and structures of networks that are linking global and local scales transferring the standard developed in the node to the sites of implementation.  The innovative approach of this paper is in the analysis of different GGNs contributing to the same site of implementation.  They turned out to be drivers and promoters of MSC-GGN, involving those that were initiated by state actors and inter-state institutions. The paper is the first in analyzing how GGNs in marine governance intersect with one another, what is their facilitating role, how they contribute to regulation of fishing and what is the added value of MSC requirements.  This approach makes this paper a state of art.

This theoretical phenomenological approach assumes qualitative methodology, in the results we look at the regulatory tools via the GGN lens. In our discussion and conclusions, we reflect on the phenomenon of several GGNs operating in the Baltic and Barents Sea and their facilitating role.

Comments to the Authors

There are a lot of typing mistakes which have been found in this revised paper.

Response: We corrected typing mistakes in the paper.

Thank you again very much for your thorough assessment!

This manuscript is a resubmission of an earlier submission. The following is a list of the peer review reports and author responses from that submission.

Round 1

Reviewer 1 Report

The article presents an analysis of the certification process in two contexts, being a relevant topic for the understanding of market-based conservation strategies in the context of fisheries.

Throughout the manuscript I made several comments to improve the quality of the publication. The main criticism is related to the discussion, which does not exist. The discussion topic should be used as a result, leaving the discussion dedicated to answering the research questions listed in lines 418-422.

I also suggest modifying the structure, such as putting a short introduction to the article before getting into the specifics of MSC. In the methods, it would be desirable to present a map of the study area and in the results some more worked product (table or graph) with the systematization of the information.

Reviewer 2 Report

Please find my comments in the attachment.

Reviewer 3 Report

The marine stewardship council (MSC) has been investigated in this research to evaluate the seafood resources through the quantitative methodologies as well as the questionable investigation for some interviewers. The cases of study have been selected in the Baltic sea (Finland) and the Barest sea (Russia). This research is interesting however there are some drawbacks that the authors should address them to improve their research paper.

  1. The structure of this research paper should be shortened with sufficient content.
  2. The main contribution of this research must be indicated clearly and analyzed intensively.
  3. In the structure of MSC, lines 109-112, “Board Committee, however, give feedback and advice to the BoT, whereas Subsidiary Boards make up separate legal entities for different key jurisdiction purposes.” should be re-written clearly.
  4. In the introduction, the authors have not stated clearly the proposed methodologies including the governance generating network theory.
  5. The position of figure 1 must be re-moved into the appropriate space.
  6. The authors only provided two cases of study including the Russia sea and Finland sea for utilizing the proposed methods. However, the quantitative analysis as well as estimating the performance of marine stewardship council has not been investigated.

Reviewer 4 Report

Line 272 to 275: the authors write following paragraph: In this paper we will access the special characteristics and compare institutionalization of MSC in Finland and Russia. We will analyze and assess whether there is an added value of MSC certification to national and international regulatory tools in either country. How do you analyze and judge about validation of results?

The index and indicators for assessing the sustainability, ecological services of sea and water quality status must describe.

The logical framework for analysis of driving force and state and response interaction is necessary. The following works are good sample for systematic analysis: Integrated groundwater management using a comprehensive conceptual framework, Flood assessment in the context of sustainable development using the DPSIR framework. You can also improve literature by them.

Pages 3 and 4, in the literature review, the introduction section must improve. You should add papers related to study and discuss about innovation of the work.

The structures of the paper are not suitable. You have important part such as: Abstract, Introduction, Material and methods (or Methodology), Results and discussion and conclusion sections in your paper. Please reorganize the paper based on these sections.

In conclusion section general comments based on research results must discuss.